# Geometry-Aware Cardiac MRI Representation Learning with Equivariant Neural Fields

**Jesse L. Wiers**[1]                                JESSE.WIERS@STUDENT.UVA.NL
**David R. Wessels**[1]                                   D.R.WESSELS@UVA.NL
**Lukas P.A. Arts**[2]                              L.P.A.ARTS@AMSTERDAMUMC.NL
**Samuel Ruiperez-Campillo**[3,4]           SAMUEL.RUIPEREZCAMPILLO@INF.ETHZ.CH
**Maarten Z.H. Kolk**[2]                            M.KOLK@AMSTERDAMUMC.NL
**Fleur V.Y. Tjong**[2*]                           F.V.TJONG@AMSTERDAMUMC.NL
**Erik J. Bekkers**[1*]                                   E.J.BEKKERS@UVA.NL

[1] *Informatics Institute, University of Amsterdam, Amsterdam, The Netherlands*

[2] *Dept. of Clinical and Experimental Cardiology, Amsterdam UMC, Amsterdam, The Netherlands*

[3] *Dept. of Computer Science, ETH Zürich, Zürich, Switzerland*

[4] *Institute of Medical Engineering and Science, MIT, Cambridge, USA*

[*] *These authors contributed equally to this work*

**Editors:** Accepted for publication at MIDL 2026

## Abstract

Cardiac MRI encodes detailed geometric information, but standard deep learning models rely on grid-based encoders that emphasize texture rather than structure. Neural fields offer a continuous alternative, yet Conditional Neural Fields (CNFs) compress each subject into a single global latent, discarding spatial organization. We evaluate Equivariant Neural Fields (ENFs) for cardiac MRI, which replace the global latent with a geometry-aware latent point cloud. ENFs achieve competitive reconstruction quality with far fewer decoder parameters and produce latents that are local, anatomically meaningful, and robust to geometric transformations. For downstream prediction tasks, ENF latents perform competitively with ResNet50 and global CNF latents across several clinical endpoints. These results position ENFs as a compact, interpretable, and geometry-aware alternative for cardiac MRI representation learning.

**Keywords:** Geometric Deep Learning, Equivariant Neural Fields, Cardiac Risk Stratification

**Code available here**

## 1. Introduction

Sudden cardiac death (SCD) remains a major global health challenge, accounting for an estimated 4–5 million deaths per year worldwide (Chugh et al., 2008). In about 50% of the cases, SCD occurs as the first clinical sign of heart disease, meaning individuals often have no prior diagnosis or recognizable symptoms before a fatal event (Adabag et al., 2010). Implantable cardioverter defibrillators (ICDs) can prevent SCD, yet current risk

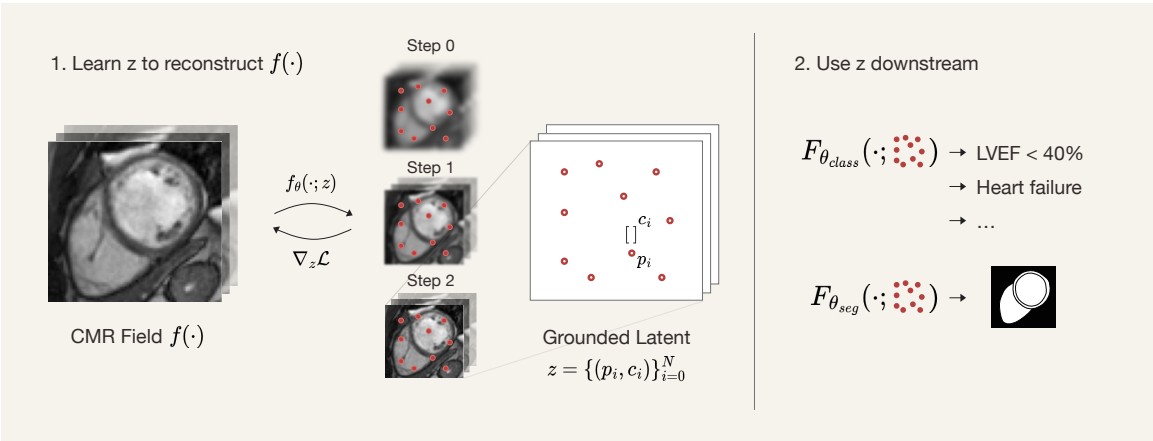

Figure 1: Equivariant Neural Fields (ENFs) ground Neural Fields (NeFs) in geometry by providing an optimized latent point cloud $z$ consisting of tuples $(p_i, c_i)$ of pose $p_i$ and context $c_i$ information. Due to their explicit positional grounding and locality, the $z$ retains geometric features in the input field that can be used for various downstream tasks (e.g., classification and segmentation).

stratification methods lack precision. This results in overtreatment, around 75% of ICD recipients never require therapy, and undertreatment, as many high-risk individuals go unidentified (Nanthakumar et al., 2004). More discriminative and robust representations of cardiac structure and function are therefore needed.

Cardiac Magnetic Resonance (CMR) imaging captures rich geometric information, including ventricular shape, motion patterns, and tissue abnormalities, which are closely tied to arrhythmic risk (Balaban et al., 2022). While CMR imaging provides rich structural information, contemporary deep learning models tend to emphasize textural cues rather than spatial structure. This phenomenon, known as texture bias, has been shown to weaken generalization and robustness (Geirhos et al., 2018; Hermann et al., 2020). Because cardiac abnormalities often manifest in subtle geometric changes, such as scar formation, wall thinning, or shape deformations of the ventricles, a representation that more explicitly encodes these geometries may offer improved diagnostic utility.

Neural fields (NeFs) have recently emerged as a paradigm for representing signals as continuous functions parameterized by coordinate-based neural networks (Xie et al., 2022). Instead of storing data on a discrete grid, a neural field learns a mapping from spatial or spatiotemporal coordinates to signal values, enabling resolution-independent reconstruction and continuous interpolation across space and time. Conditional Neural Fields (CNFs) extend this idea by modulating a shared neural field with per-sample latent vectors, allowing the same decoder to reconstruct many individual subjects (Dupont et al., 2022; Friedrich et al., 2025). While CNFs achieve strong reconstruction quality, their *global* latent vectors discard local spatial structure—limiting geometric interpretability and the ability to reason about localized pathology.

Equivariant Neural Fields (ENFs) have recently addressed this limitation by replacing the global latent with a *latent point cloud* comprising pose–appearance tuples (Wessels et al., 2024) (see Figure 1). This design grounds the representation in geometry, ensuring (i) **locality**, as each latent encodes a specific spatial region, and (ii) **equivariance**, meaning transformation groups such as rotations or translations can induce predictable transformations in the latent space. These properties yield spatially structured, transformation-aware representations that are well suited for medical data, where acquisition pose can vary and clinically relevant features are often local.

**Contributions.** This work provides the first application and systematic evaluation of Equivariant Neural Fields (ENFs) on a medical imaging modality, specifically an evaluation of ENFs for cardiac MRI representation learning and risk stratification. Our main contributions are:

1. **Improved CMR reconstruction over CNFs.** We show that ENFs achieve equal or on par reconstruction fidelity compared to CNFs for both 2D and 4D cine CMR, while using substantially fewer decoder parameters.

2. **Geometric advantages of ENFs in cardiac MRI.** We empirically demonstrate that ENF latent point clouds exhibit spatial locality and transformation stability in a clinical CMR setting—properties absent in CNF and Convolutional Neural Network (CNN) representations.

3. **ENFs as a competitive alternative for clinical prediction.** We show that ENF latent representations achieve risk prediction accuracy on par with, and often exceeding, CNF global latents and grid-based image models.

## 2. Related Work

### 2.1. Conditional Neural Fields

Neural fields represent signals as continuous functions $f_\theta : x \mapsto y$ parameterized by coordinate-based neural networks (Xie et al., 2022), enabling resolution-independent reconstruction and memory-efficient storage. Conditional Neural Fields (CNFs) extend this idea by conditioning the shared decoder on a latent variable specific to each signal (Dupont et al., 2022). Formally, a CNF models:

$$f_\theta(x; z_i) : \mathbb{R}^d \times \mathbb{R}^k \to \mathbb{R}^c, \tag{1}$$

where $x$ is a spatial or spatiotemporal coordinate and $z_i$ is a per-instance latent vector that captures subject-specific variations. This enables a single decoder to represent an entire dataset, generalizing classical neural fields that require training a separate network per sample.

Latent variables $z_i$ are typically learned through either (i) *autodecoding*, where $z_i$ and $\theta$ are jointly optimized using gradient descent (Park et al., 2019), or (ii) *model-agnostic meta-learning* (MAML), where the neural field parameters $\theta$ are meta-learned to make latent optimization efficient. At test time, each latent vector $z_i$ is randomly initialized and is refined with only a few gradient steps on the reconstruction loss, enabling fast adaptation to new signals (Tancik et al., 2021). Pseudo algorithms for both learning strategies can be

found in Appendix A.

CNFs are commonly instantiated using sinusoidal networks such as SIREN (Sitzmann et al., 2020), which address spectral bias, allowing to represent high-frequency details. CNFs have been shown to be able to accurately reconstruct a broad range of heterogeneous medical modalities, from ECG waveforms to 2D fundus images and 3D/4D MRI volumes (Friedrich et al., 2025). Despite their strong reconstruction fidelity, CNFs encode each signal using a single global latent vector that lacks spatial grounding—it captures what features exist but not where. As a result, global latents cannot capture locality, pose, or geometry, making CNFs sensitive to acquisition variation and limiting their usefulness for cardiac MRI. These limitations motivate geometry-aware extensions such as Equivariant Neural Fields (ENFs), which replace the global latent with a structured latent point cloud enforcing group-theoretic constraints (Wessels et al., 2024).

## 3. Method

### 3.1. Equivariant Neural Fields (ENFs)

Equivariant Neural Fields (ENFs) extend Conditional Neural Fields (CNFs) by replacing the global latent vector with a *geometric* latent point cloud:

$$z = \{(p_i, c_i)\}_{i=1}^K, \tag{2}$$

where $p_i \in G$ denotes a pose in a transformation group $G$ (e.g. $SE(2)$) and $c_i \in \mathbb{R}^{d_c}$ is a local appearance vector. A group element $g \in G$ acts on the input field via $(L_g f)(x) := f(g^{-1}x)$, and on the latent point cloud via $g \cdot z = \{(gp_i, c_i)\}_{i=1}^K$. ENFs are constructed to satisfy the *steerability* condition:

$$f_\theta(g^{-1}x; z) = f_\theta(x; g \cdot z), \tag{3}$$

meaning that geometric transformations of the input correspond to structured transformations in latent space. This provides a geometry-aware representation that preserves pose information and separates geometry (through $p_i$) from appearance (through $c_i$).

To enforce steerability, the decoder must depend on a *bi-invariant* relation between coordinates and latent poses. ENFs encode this relation through a relative attribute:

$$a(x, p_i) := \varphi(p_i^{-1}x), \tag{4}$$

where $\varphi$ is a Gaussian Fourier embedding (Tancik et al., 2020). Because $a(x, p_i)$ depends only on $p_i^{-1}x$, transforming both $x$ and $p_i$ by any $g \in G$ leaves the attribute unchanged, making the decoder bi-invariant and guaranteeing (3). A full description of the bi-invariant attribute parameterizations used in this work is provided in Appendix C. ENFs predict values at coordinates $x_m$ by attending to the latent point cloud:

$$f_\theta(x_m; z) = W_o \sum_{i=1}^K \text{att}(x_m, z_i) \, \mathbf{v}(a(x_m, p_i), c_i), \tag{5}$$

where the attention operation is:

$$\text{att}(x_m, z_i) = \text{softmax}_i\left(\frac{\mathbf{q}(a(x_m, p_i))^\top \mathbf{k}(c_i)}{\sqrt{d_k}} + \mu_\sigma(x_m, p_i)\right). \tag{6}$$

The parameterizations of the cross-attention terms can be found in Appendix B. The term $\mu_\sigma(x_m, p_i) = -\sigma_{\text{att}} \|x_m - p_{\text{pos},i}\|^2$ encourages locality by down-weighting latents that are spatially distant from the coordinate and effectively induces a Gaussian windowing through the softmax operator.

The full cross-attention in (6) scales as $\mathcal{O}(KM)$, where $K$ is the number of latents and $M$ the number of coordinates. To reduce computational cost while maintaining locality, attention is computed only over the $k$ nearest latent poses for each coordinate $x_m$, resulting in a $k$-nearest-neighbour (kNN) approximation. This preserves the geometric structure of the ENF while enabling efficient reconstruction of high-resolution CMR data. The latent point cloud $z$ can be obtained using either (i) *model-agnostic meta-learning* (MAML) or (ii) *autodecoding* (More details on the learning algorithms are provided in Section 2.1 and Appendix A).

## 3.2. Dataset and Preprocessing

To evaluate both reconstruction fidelity and downstream clinical prediction performance, cardiac cine magnetic resonance imaging (CMR) and outcome labels from the UK Biobank (UKB), are used (Sudlow et al., 2015). The UKB provides one of the largest publicly available CMR cohorts, containing over 40,000 studies acquired under a standardized imaging protocol. Each study contains short-axis (SAX) cine volumes covering the full cardiac cycle, with a typical in-plane resolution of $1.8 \times 1.8\,\text{mm}$, a slice thickness of $8\,\text{mm}$, and 50 temporal frames. These volumes include the left and right ventricles as well as the surrounding myocardium.

The UK Biobank contains a wide range of clinical endpoints; in this work, we focus on left ventricular ejection fraction (LVEF), cardiomyopathy, sudden cardiac death, all-cause mortality, heart failure, myocardial infarction, atrial fibrillation and ischaemic heart disease. The exact number of patients associated with each endpoint and the UK Biobank ICD codes used to define them are provided in Appendix D. After accounting for diagnostic overlap, the total number of unique patients with at least one of these conditions is 7,121. To construct a balanced cohort for all reconstruction and downstream prediction experiments, we sample an equal number of healthy participants without any of the target diagnoses, yielding a final dataset of 14,242 subjects.

To isolate the heart and remove surrounding background from each CMR image, we use the publicly released UKB segmentation model of Bai et al. (Bai, 2018) to obtain left-ventricular (LV) cavity, LV myocardium, and right-ventricular (RV) cavity segmentations for each short-axis slice. A tight axis-aligned bounding box is computed around the union of these three masks and expanded to a standardized size derived from the population-wide distribution of bounding box dimensions. This crop is applied uniformly across all slices and frames of a study, ensuring consistent and anatomically aligned spatial coverage of the cardiac region. An illustration of the preprocessing is displayed in Figure 2.

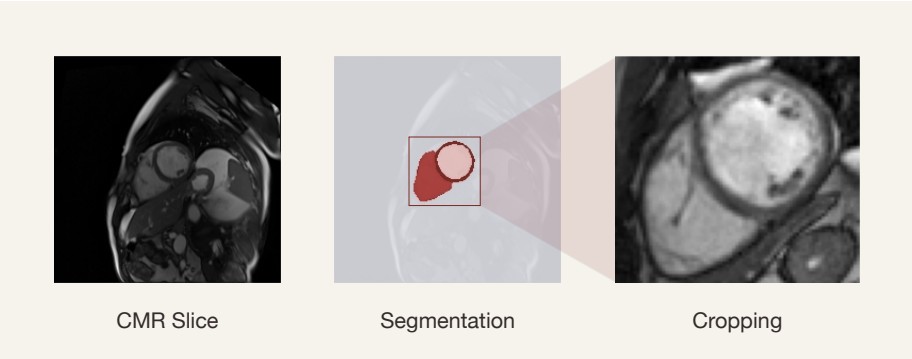

Figure 2: CMR preprocessing pipeline: (1) identify the slice with maximal myocardium area; (2) compute and standardize a bounding box; (3) crop all slices accordingly. Adapted from the preprocessing workflow in the UKB CMR pipeline.

## 4. Experiments and Results

### 4.1. Reconstruction of CMR Images

We assess the capacity of ENFs to reconstruct CMR images under different latent resolutions and compare them against CNF baselines matched for total latent dimensionality. Models are either trained on spatial coordinates of 2D CMR slices or spatio-temporal coordinates of 4D cine volumes. All experiments are conducted on the full dataset described in Section 3.2, using an 80/10/10 train–validation–test split. Reconstruction targets are voxel intensities, learned through a meta-learning procedure in which a small number of inner-loop updates adapt the latent variables for each patient. Following the sampling strategy used in MedFuncta (Friedrich et al., 2025), each forward pass observes a random subset of 25% of the available coordinates. Reconstruction accuracy is quantified using the peak signal-to-noise ratio (PSNR) on held-out test slices and volumes. All ENF and CNF models are trained for 25,000 iterations with a batch size of 16; a full list of hyperparameters is provided in Appendix E. Both ENF and CNF models are trained using identical data splits, coordinate subsampling strategies, reconstruction targets, and evaluation protocols; differences are limited to the latent parameterization and architecture.

Table 1 reports the PSNR scores for the three latent resolutions and the different dimensionalities. When reconstructing 2D slices, ENFs substantially exceed the performance of CNFs at higher latent capacities (64×32 and 128×64) despite using an order of magnitude less decoder parameters. When reconstructing 4D volumes, ENFs obtain similar performance scores across all latent resolutions. However, the absolute PSNR values for both ENF and CNF remain low (maximum of ≈20 dB), indicating that neither model is able to capture the spatiotemporal complexity of the 4D volume using meta-learning. This suggests that 4D reconstruction is substantially harder and likely requires either stronger decoders or alternative training strategies (e.g., autodecoding) to reach clinically meaningful fidelity. Initial exploratory results using autodecoding for the smallest latent configuration (32 latents of size 16) indicate improved 4D reconstruction fidelity for ENFs (25.73 dB PSNR), suggesting that the low PSNR under meta-learning is largely attributable to the training regime.

Table 1: Reconstruction PSNR (dB) and decoder parameter count (in millions) for ENF and CNF models at matched total latent dimensionalities. Latent sizes correspond to ENF using $K \times d$ pose–appearance latents and CNF using a global latent vector of size $K \times d$.

| Model | 32×16 | 64×32 | 128×64 |
|---|---|---|---|
| **ENF (2D CMR)** | 24.21 (0.743M) | **34.31** (0.760M) | **39.62** (0.793M) |
| **CNF (2D CMR)** | **26.22** (2.56M) | 29.98 (7.68M) | 32.02 (28.12M) |
| **ENF (4D CMR)** | 17.65 (0.743M) | 19.12 (0.760M) | 19.80 (0.793M) |
| **CNF (4D CMR)** | **20.22 (2.56M)** | **20.44 (7.68M)** | **20.51 (28.12M)** |

However, the primary focus of this work is not on achieving state-of-the-art reconstruction performance, but on learning geometry-aware representations with strong downstream utility.

## 4.2. Downstream Prediction

To assess the clinical usefulness of ENF-derived latent point clouds, we trained 7 classification models for various latent configurations, targeting clinically relevant endpoints: cardiomyopathy, sudden cardiac death, all-cause mortality, heart failure, myocardial infarction, atrial fibrillation, and ischaemic heart disease. Each model was trained for 100 epochs using an 80/10/10 train–validation–test split. All ENF models, unless otherwise stated, use the translational bi-invariant, which showed the strongest downstream performance in Wessels et al. (2024). An overview of the hyperparameters used for all downstream experiments is provided in Appendix E.

**Effect of latent resolution.** To assess how the structure of the ENF latent grid affects downstream performance, we evaluate three configurations—$32 \times 16$, $64 \times 32$, and $128 \times 64$—where the first value specifies the number of latent vectors and the second the dimensionality of each latent vector. For each patient, latent point clouds are extracted from the middle three short-axis slices in the z-dimension across multiple cardiac phases ($T \in \{0, 10, 20, 40, 49\}$) using 2D reconstruction. These latents are then stacked into a fixed-size point cloud and passed to a transformer-based classifier. Table 2 summarizes the performance for all three latent configurations.

Overall, increasing the latent grid size does not consistently improve downstream accuracy. Although larger configurations yield higher reconstruction fidelity (Section 4.1), this does not directly translate into more discriminative representations for clinical endpoints. This suggests that visually sharper reconstructions do not necessarily encode more task-relevant information. Smaller latent clouds may already capture the geometric features required for downstream prediction. However, we note that smaller configurations exhibit higher variance across runs, indicating reduced stability compared to the larger latent grids.

**Comparison to baseline representations.** We further compare ENF-derived latents to two baseline representations: i) 2D image slices processed with a ResNet50 classifier, and ii) global latent vectors obtained from a SIREN-based CNF. ENF latents and ResNet50

Table 2: Endpoint prediction accuracy for different ENF latent resolutions (MAML latents; accuracy mean ± SD over 3 runs).

| Endpoint | 32×16 (PSNR = 24.21) | 64×32 (PSNR = 34.31) | 128×64 (PSNR = 39.62) |
|---|---|---|---|
| Cardiomyopathy | 0.60 (± 0.07) | 0.58 (± 0.09) | **0.66 (± 0.01)** |
| Sudden Cardiac Death | **0.64 (± 0.09)** | 0.48 (± 0.05) | 0.48 (± 0.02) |
| All-Cause Mortality | 0.59 (± 0.02) | 0.59 (± 0.03) | **0.63 (± 0.03)** |
| Heart Failure | 0.63 (± 0.05) | **0.65 (± 0.07)** | 0.64 (± 0.05) |
| Myocardial Infarction | 0.58 (± 0.04) | **0.62 (± 0.01)** | 0.60 (± 0.03) |
| Atrial Fibrillation | **0.62 (± 0.01)** | 0.61 (± 0.03) | **0.62 (± 0.01)** |
| Ischaemic Heart Disease | **0.59 (± 0.02)** | **0.59 (± 0.02)** | 0.58 (± 0.01) |

Table 3: Endpoint prediction accuracy for different representations (mean ± SD over 3 runs).

| Endpoint | ENF (MAML) + Transformer | ResNet50 | SIREN + MLP |
|---|---|---|---|
| Cardiomyopathy | 0.60 (± 0.07) | 0.61 (± 0.10) | **0.62 (± 0.07)** |
| Sudden Cardiac Death | **0.64 (± 0.09)** | 0.49 (± 0.10) | 0.57 (± 0.08) |
| All-Cause Mortality | **0.59 (± 0.02)** | 0.48 (± 0.03) | 0.53 (± 0.07) |
| Heart Failure | 0.63 (± 0.05) | 0.56 (± 0.05) | **0.64 (± 0.07)** |
| Myocardial Infarction | 0.58 (± 0.04) | **0.59 (± 0.02)** | 0.57 (± 0.04) |
| Atrial Fibrillation | **0.62 (± 0.01)** | 0.60 (± 0.02) | 0.58 (± 0.03) |
| Ischaemic Heart Disease | **0.59 (± 0.02)** | 0.55 (± 0.00) | 0.57 (± 0.02) |

models were trained on the three central short-axis slices (z-dimension) extracted at multiple cardiac timepoints, $T \in \{0, 10, 20, 40, 49\}$. In contrast, the CNF model produces a single global latent vector per patient, obtained by conditioning on the full 4D cine volume during reconstruction rather than on individual slices. These global latents are provided to a simple residual MLP classifier. Table 3 summarizes the comparative performance across the three representations.

Across endpoints, no single representation dominates. ENF latents achieve the highest accuracy on four tasks (i.e., sudden cardiac death, all-cause mortality, atrial fibrillation, and ischaemic heart disease), showing that they provide a competitive representation despite being learned only through reconstruction. ResNet50 performs strongest on myocardial infarction, while the SIREN–MLP baseline performs best on cardiomyopathy and heart failure, indicating that global latents can still be effective for some endpoints. The relatively large standard deviations across all methods highlight substantial variability, suggesting that these clinical prediction tasks are inherently challenging. Overall, the results show that ENFs perform competitively with both baselines, offering complementary strengths depending on the endpoint. Notably, it can be observed that increasing the latent capacity does not consistently translate into improved downstream performance. Given the addi-

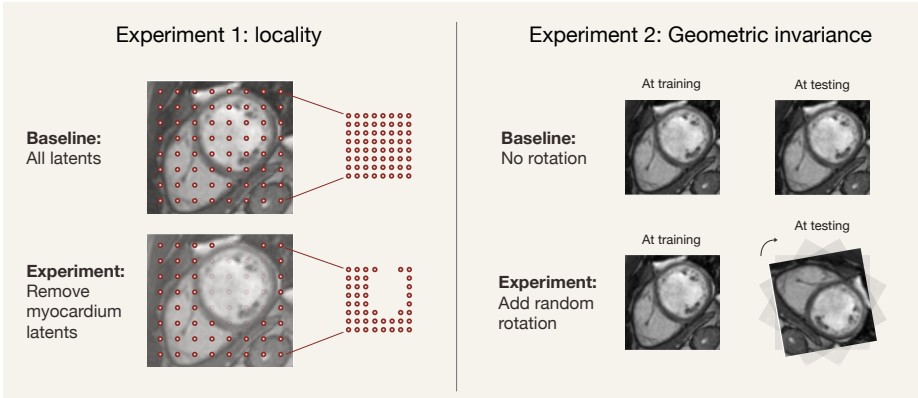

Figure 3: We conducted two ablation experiments to test ENF's locality and geometric invariance. Locality was evaluated by measuring the performance drop after removing the latents closest to the myocardium center. Geometric invariance was evaluated by rotating each test input by a random angle (90°, 180°, or 270°) to assess the stability of the learned representation under in-plane rotations.

tional computational cost associated with larger latent configurations, these results suggest that training ENFs with relatively compact latent point clouds is preferable in practice, as they achieve competitive performance while maintaining efficiency.

### 4.3. Geometric Properties of ENF Latents

ENFs generate latent point clouds that are both spatially meaningful and stable under geometric transformations. In this section, we present two ablation studies that evaluate two core properties of ENFs: *locality* and *invariance to geometric transformations* (see Figure 3). Finally, we assess their robustness to variations in scanner type and patient cohort using a clinically relevant group of patients diagnosed with dilated cardiomyopathy (DCM).

**Locality** To evaluate whether ENF latents capture spatially local and functionally meaningful cardiac information, we conduct an ablation study using a binary LVEF prediction task. We first extract latent point clouds with MAML (64 latents per slice, 32-dimensional latents) from the middle three short-axis slices at end-diastolic (ED) and end-systolic (ES) frames. We then remove the $k = 16$ latents closest to the anatomical center of the myocardium, ranked by Euclidean distance (see Figure 3).

When comparing LVEF prediction performance with and without the myocardium-adjacent latent points, we observe a 28% drop in accuracy (from $86 \pm 0.01\%$ to $58 \pm 0.04\%$).Without these points, model performance collapses toward random guessing, confirming that latents near the ventricular wall encode critical clinical information. This spatial organization is further validated by the model's gradient-based saliency map (Figure 4), which demonstrates that the ENF latent space is both anatomically meaningful and highly structured.

**Robustness to geometric transformations** To evaluate whether ENFs produce representations that are stable under geometric perturbations, we test their behavior under in-plane rotations of the input CMR slices. Standard CNN-based encoders often degrade

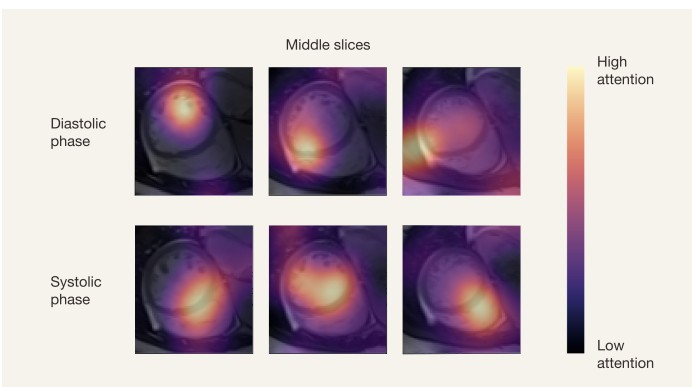

Figure 4: Gradient-based saliency map showing areas of high and low importance in the task of low LVEF classification based on diastolic and systolic MRI slices for a subject with LVEF=17%. As expected, the heatmaps reveal a strong attention to the myocardium wall.

Table 4: Accuracy on original vs. rotated test sets (mean ± SD).

| Model | Original Accuracy | Rotated Accuracy |
|---|---|---|
| ResNet50 (Image Slices) | **0.93 (± 0.01)** | 0.85 (± 0.02) |
| ENF (MAML) | 0.86 (± 0.01) | **0.86 (± 0.01)** |

when exposed to rotations not seen during training, due to their dependence on image-grid coordinates. In contrast, ENFs encode information in a spatially grounded, transformation-consistent manner and maintain performance under such changes. We use the same latent configuration as in the locality analysis (64 latents per slice, latent dimension 32, obtained via MAML). Latents are extracted from the three central short-axis slices at end-diastolic (ED) and end-systolic (ES) frames. For this experiment, the ENF is trained using an $SE(2)$ roto-translation bi-invariant, encouraging representations that are consistent under in-plane rotations and translations. After training, both ENF-derived latent representations and image-slice inputs for the ResNet50 baseline are evaluated on a rotated version of the test set. Each slice is rotated by a random angle chosen from $\{90°, 180°, 270°\}$ (see Figure 3).

As shown in Table 4, ENF-based LVEF prediction is unchanged under rotation (0.86 → 0.86), whereas the ResNet50 baseline drops substantially (0.93 → 0.85). These results indicate that ENF latent point clouds encode cardiac anatomy in a geometrically robust manner, maintaining predictive performance even under significant acquisition-angle variation.

**Robustness to data shift** We hypothesize that ENF's locality and geometric invariance properties ensure robust performance regardless of patient cohort, scanner type, and pathology. As the UK Biobank predominantly comprises healthy subjects, we set out to validate ENF's reconstruction capability on a different, clinically relevant patient cohort. This dataset includes 978 patients diagnosed with Dilated Cardiomyopathy (DCM) that

were referred for genetic testing at Amsterdam UMC. In comparison, only 0.11% of subjects in the UK biobank were diagnosed with DCM. A 4D short-axis CMR was recorded for each patient. Similar to the previous experiments, CMR was first segmented and cropped. Subsequently, three latent point cloud configurations (i.e., $32 \times 16$, $64 \times 32$, and $128 \times 64$) were created per frame, per slice.

Using identical hyperparameters and training for 2,500 iterations with a batch size of 4, we obtained reconstruction PSNRs of 25.76 dB, 29.40 dB, and 32.09 dB for the three latent configurations, respectively. Notably, the smallest configuration achieved higher reconstruction performance than reported for the UK Biobank. The two larger configurations further improved reconstruction quality, although still falling short of UK Biobank performance. This is likely caused by the substantially smaller training set (i.e., our dataset includes $40\times$ fewer patients). Importantly, all three configurations remain viable, as downstream classification performance was only marginally affected by the size of the latent point cloud.

## 5. Conclusion and Discussion

Equivariant Neural Fields provide a geometry-aware alternative for cardiac MRI representation learning, offering competitive reconstruction and predictive performance while producing interpretable, spatially grounded latents. The results highlight the potential of ENFs as a robust foundation for downstream clinical tasks. Several limitations remain however. Our evaluation has been limited to short-axis CMR only. Additionally, while meta-learning encourages geometric structure, it constrains reconstruction fidelity for high-dimensional 4D cine data. Future work could explore full $SE(3)$-equivariant ENFs, alternative training strategies beyond meta-learning, and broader evaluations, both by using additional, structure-based reconstruction metrics (e.g., structural similarity index) and across a wider range of downstream tasks using multi-center datasets. Another promising direction is multimodal ENFs, where a shared latent point cloud jointly represents CMR together with complementary signals such as ECG, potentially capturing richer cardiac structure and physiology within a unified geometric framework.

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

## Appendix A. Learning the Latent Representation

CNFs and ENFs rely on a latent variable $z$ that is optimized jointly with the shared decoder. In this work, we use two standard strategies to learn these latents: *autodecoding*, where each latent vector is directly optimized for every subject, and *model-agnostic meta-learning* (MAML), which learns an initialization that can be adapted with a few inner-loop updates. For completeness, Algorithms 1 and 2 provide the procedures used to obtain the latent representations in our reconstruction and downstream experiments.

---

**Algorithm 1** MAML

---

Randomly initialize shared neural field $f_\theta$
**for** number of training iterations **do**
    Sample batch of signals $f$
    Sample random coordinates $x$
    Initialize latent vectors $z_f^{(0)}$ for all $f \in \mathcal{B}$
    **for** number of inner loop steps **do**
        Compute reconstruction loss $\mathcal{L}_{\mathrm{MSE}}(f_\theta(x, z_f^{(t)}), f(x))$
        Update latent vectors:

$$z_f^{(t+1)} \leftarrow z_f^{(t)} - \epsilon \nabla_{z_f^{(t)}} \mathcal{L}_{\mathrm{MSE}} \quad \forall f \in \mathcal{B}$$

    **end for**
    Compute reconstruction loss across batch:

$$\mathcal{L}_{\mathrm{MSE}}^{\mathrm{batch}} = \frac{1}{|\mathcal{B}|} \sum_{f \in \mathcal{B}} \|f_\theta(x, z_f^{(t+1)}) - f(x)\|^2$$

    Update neural field parameters:

$$\theta^{(t+1)} \leftarrow \theta^{(t)} - \eta \nabla_{\theta^{(t)}} \mathcal{L}_{\mathrm{MSE}}^{\mathrm{batch}}$$

**end for**

---

---

**Algorithm 2** Autodecoding

Randomly initialize shared neural field $f_\theta$
Initialize latent vectors $z_f^{(0)}$ for all signals $f \in \{1, \ldots, N\}$
**for** Number of training iterations **do**
    Sample batch of signals $f$
    Sample random coordinates $x$
    Compute reconstruction loss $\mathcal{L}_{\text{MSE}}(f_\theta(x, z_f^{(t)}), f(x))$
    Update latent vectors and neural field parameters:

$$z_f^{(t+1)} \leftarrow z_f^{(t)} - \epsilon \nabla_{z_f^{(t)}} \mathcal{L}_{\text{MSE}}$$

$$\theta^{(t+1)} \leftarrow \theta^{(t)} - \eta \nabla_{\theta^{(t)}} \mathcal{L}_{\text{MSE}}$$

**end for**

---

## Appendix B. Cross-Attention Parameterization

For completeness, we provide the parameterization of the components used in the ENF cross-attention mechanism. Given a coordinate–latent pair $(x, p_i, c_i)$, the invariant pairwise attribute and the query, key, and value parameterizations used in the attention operation are defined as:

$$\mathbf{a}(x, p_i) := \varphi(p_i^{-1} x), \tag{7}$$

$$\mathbf{q}(\mathbf{a}) := W_q \, \mathbf{a}, \tag{8}$$

$$\mathbf{k}(c_i) := W_k \, c_i, \tag{9}$$

$$\mathbf{v}(\mathbf{a}, c_i) := (W_v c_i) \odot (W_{a\gamma} \mathbf{a}) + (W_{a\beta} \mathbf{a}), \tag{10}$$

where $\varphi(\cdot)$ is an RFF embedding and $\odot$ denotes element-wise multiplication.

## Appendix C. Bi-Invariant Function Parameterizations

The bi-invariant attributes used in the ENF experiments are detailed below. These parameterizations ensure steerability by encoding geometric relations that are invariant to specific transformation groups.

TRANSLATIONAL SYMMETRIES $\mathbb{R}^n$

In this setting, poses correspond to translations $t_i \in \mathbb{R}^n$. The bi-invariant is defined as:

$$a_{m,i}^{\mathbb{R}^n} = x_m - t_i \tag{11}$$

ROTO-TRANSLATIONAL SYMMETRIES $SE(2)$

For the group $SE(2)$, which models planar rotations and translations, poses $p_i$ correspond to group elements $g = (\theta_i, t_i) \in SE(2)$. We use the invariant attribute proposed by Bekkers et al. (2023):

$$a_{m,i}^{SE(2)} = R_{\theta_i}(x_m - t_i) \tag{12}$$

where $R_{\theta_i}$ denotes the 2D rotation matrix corresponding to angle $\theta_i$.

## Appendix D. Patient Counts Per Clinical Endpoint

Table 5 reports the number of patients associated with each clinical endpoint considered in this work, along with the corresponding ICD-10 codes used in the UK Biobank dataset.

| Condition | Patients | ICD-10 Codes |
|---|---|---|
| Cardiomyopathy | 237 | I420–I429 |
| Sudden Cardiac Death | 237 | I460, I461, I469, I472, I490 |
| All-Cause Mortality | 712 | N/A |
| Heart Failure | 823 | I500–I509 |
| Myocardial Infarction | 1421 | I210–I229 |
| Atrial Fibrillation | 2210 | I480, I481, I482, I489 |
| Ischaemic Heart Disease | 4729 | I200–I259 |

Table 5: Number of patients per clinical endpoint considered in this work.

## Appendix E. Additional Experiment Details

This appendix lists the full set of hyperparameters used for all the experiments and provides additional details regarding methods used.

### E.1. Reconstruction Experiments

#### E.1.1. ENF (META-LEARNING)

Hyperparameters used to train the ENF models:

| Hyperparameter | Value |
|---|---|
| Global learning rate | $5 \times 10^{-4}$ |
| Total training iterations | 25,000 |
| Batch size | 16 |
| Hidden dimensionality | 256 |
| Number of attention heads | 3 |
| Attention head dimensionality | 256 |
| $k$-nearest neighbours | $k = 4$ |
| RFF std. $(\sigma_q, \sigma_v)$ | (30.0, 60.0) |
| Latent learning rates (pos., ctx.) | (0.0, 60.0) |

Table 6: Hyperparameters used for ENF meta-learning reconstruction experiments.

### E.1.2. CNF (Meta-Learning)

Hyperparameters used to train the CNF models:

| Hyperparameter | Value |
|---|---|
| Depth (number of layers) | 15 |
| Hidden dimensionality | 256 |
| Batch size | 16 |
| Context selection ratio | 0.25 |
| Sinusoidal frequencies ($\omega_1$, $\omega_K$) | (20, 200) |
| Inner-loop learning rate | $1 \times 10^{-2}$ |
| Outer-loop learning rate | $3 \times 10^{-6}$ |

Table 7: Hyperparameters used for CNF meta-learning reconstruction experiments.

### E.2. Downstream performance

### E.2.1. Transformer Decoder

The downstream experiments use a lightweight Transformer-based decoder inspired by the DiT architecture (Peebles and Xie, 2023). The decoder operates directly on the latent point cloud by combining each latent point's coordinate $p_0$ with its appearance vector $c_0$. A learned positional embedding is computed from the coordinates and added to the context embedding, producing a sequence of tokens associated with the latent points.

This sequence is processed by a stack of self-attention blocks, each consisting of multi-head attention followed by a feed-forward network, with residual connections and layer normalization throughout. The feed-forward sublayer expands the hidden dimension by a factor of four before projecting back down, following the standard DiT/ViT design. After the final block, global average pooling aggregates the token representations into a single feature vector, which is mapped to class logits by a linear projection.

| Hyperparameter | Value |
|---|---|
| Hidden size | 2048 |
| Number of layers (depth) | 12 |
| Number of attention heads | 12 |
| MLP ratio (expansion factor) | 4 |
| Positional embedding | Sinusoidal (freq. scale = 1.0) |
| Learning rate | $1 \times 10^{-5}$ |

Table 8: Hyperparameters used for the Transformer decoder in downstream experiments.

### E.2.2. ResNet50

The ResNet50 model followed the standard architecture (He et al., 2016).

| Hyperparameter | Value |
|---|---|
| Bottleneck configuration | [3, 4, 6, 3] |
| Learning rate | $1 \times 10^{-5}$ |

Table 9: Hyperparameters used for the ResNet50 classifier in downstream experiments.

### E.2.3. CNF MLP

To evaluate the informativeness of the global latent vectors produced by the SIREN-based CNF model, a lightweight residual MLP classifier was used. The classifier maps each subject's global latent vector to endpoint predictions through an initial projection to a hidden representation, followed by a sequence of residual fully connected blocks. Each block applies a linear layer, LayerNorm, and GELU activation with a residual skip connection. A final linear layer maps the resulting feature vector to the target number of classes.

| Hyperparameter | Value |
|---|---|
| Hidden dimensionality | 1024 |
| Number of residual blocks | 3 |
| Learning rate | $1 \times 10^{-5}$ |

Table 10: Hyperparameters used for the CNF MLP classifier on SIREN global latent vectors.

