# OpenReview forum: "Geometry-Aware Cardiac MRI Representation Learning with Equivariant Neural Fields"
_MIDL.io/2026/Conference — MIDL 2026 Poster_

### Official Review · Reviewer_rvrd · 2026-01-08

**Confidence:** 4
**Preliminary Rating:** 4
**Final Rating:** 5

**Summary:**

The paper presents equivariant neural fields (ENFs), geometry aware representation learning framework for cardiac MRI. The idea is to replace the global latent vector used in conditional neural fields (CNFs) with a latent point cloud which is based on spatial coordinates and is equivariant to geometric transformations. They evaluate the model on many tasks such as reconstruction, downstream clinical prediction, and robustness to rotations and data shifts, compared to CNF and CNN-based models.

**Strengths:**

The paper is very well written and scoped, with multiple evaluations showing the capability of the model. The authors are transparent about the model's and work's limitations, and the work has strong reproducibility with provided code, full architecture details and hyperparameters.
The results compared to CNFs show better or similar reconstruction quality. The test on geometric transformations shows that the ENF prediction is unchanged, proving its geometrically robust features.

**Weaknesses:**

As the authors also mention, there are limitations in the performance gains over strong baselines and it's not consistent.
It is not clear in the Locality test, why they perform both the removal of latents and the addition of rotation at the same time, doing it one at a time and showing the performance could also give insights and make the points stronger.
Also, comparing endpoint prediction, the rotation test and the LVEF prediction also on CNF (since that's the NF alternative) could give insights on how much of the robustness comes specifically from the geometric latent point cloud.

**Detailed Comments:**

The choice not to compare the other tests on CNF.

**Justification Of Final Rating:**

The authors have addressed the reviewer comments, and the revised manuscript more clearly highlights the strong equivariant properties of the proposed model, and are aware of its limitations. The demonstrated equivariant behavior, solid empirical evaluation, and strong reproducibility make this a valuable and technically sound contribution.

**Justification Of The Preliminary Rating:**

It's overall a solid and well-written paper, methodologically clear and reproducible, with honest evaluation and analysis. But since the ENF doesn't consistently outperform the baselines, it seems that the only strength is its equivarience.

**Questions To Address In The Rebuttal:**

Since the performance is not consistent across latent sizes, what do they propose?

---

> ### Author Response · Authors · 2026-01-25
> **Responses to 'Questions To Address In The Rebuttal'**
>
> We thank the reviewer for their thorough assessment of our work, and are glad to see that the reviewer appreciates the clarity and proper evaluation of the proposed method. The reviewer raises a number of valid questions, which we address below.
>
> ### Questions to Address in the Rebuttal
>
> **(1) Since the performance is not consistent across latent sizes, what do the authors propose?**
>
> The observed variation across latent sizes indicates that increasing latent capacity does
> not necessarily translate into improved downstream performance. We argue that this is due to over-
> parametrization of the representation, resulting in SGD becoming able to overfit on the pixel values
> themselves instead of extracting usefull features for downstream tasks. We therefore recommend
> keeping the latent point cloud as small as possible while maintaining competitive performance, balancing representational capacity with efficiency. This reinforces one of the practical advantages of
> ENFs: strong performance can be achieved with relatively compact latent representations. We have mentioned this proposal more expliclity in the paper.
>
> ### Detailed Comments
>
> **(1) The choice not to compare the other tests on CNF.**
>
> CNFs are included as the primary neural field baseline for reconstruction and downstream endpoint prediction throughout the paper. For the locality and rotation experiments, our focus is on
> isolating properties that arise specifically from spatially grounded and equivariant latent representations. In the rotation experiment, we therefore additionally compare against a ResNet-based model, as convolutional networks with standard grid-based representations represent the commonly used
> baseline in medical image analysis.
>
> ### Weaknesses
>
> **(1) Performance gains over strong baselines are not consistent across tasks. In the locality test, it
> is unclear why latent removal and rotation are applied simultaneously rather than separately. Additionally, endpoint prediction and robustness tests are not reported for CNFs, making it difficult
> to disentangle the contribution of the geometric latent point cloud.**
>
> The locality and rotation experiments are conducted as separate tests: latent removal is used
> exclusively in the locality experiment and vice versa. With respect to baselines, CNFs are included
> throughout the paper as the primary neural field alternative and are compared against ENFs for reconstruction and downstream endpoint prediction. In the rotation experiment, we compare against
> a ResNet-based model, as convolutional networks with standard grid-based representations remain
> the prevailing approach in medical image analysis.

---

### Official Review · Reviewer_Byb5 · 2026-01-08

**Confidence:** 4
**Preliminary Rating:** 4
**Final Rating:** 5

**Summary:**

This work evaluates Equivariant Neural Fields as an alternative for cardiac MRI (CMR) representation learning and risk stratification. Unlike standard Conditional Neural Fields that use a single global latent vector, ENFs utilize a latent point cloud that preserves spatial organization and ensures equivariance to geometric transformations. The authors demonstrate that ENFs achieve competitive reconstruction fidelity with substantially fewer decoder parameters while providing interpretable, locally grounded representations.

**Strengths:**

-The work represents to my knowledge the first application and systematic evaluation of Equivariant Neural Fields (ENFs) in a medical imaging context. By replacing global latent vectors with geometric point clouds, the authors introduce a framework that inherently respects the spatial organization of cardiac anatomy.

-Parameter Efficiency: A significant technical achievement is the model's efficiency; ENFs achieve reconstruction fidelity on par with or exceeding baseline Conditional Neural Fields (CNFs) while utilizing an order of magnitude fewer decoder parameters.

-The authors empirically demonstrate that ENF representations are stable under in-plane rotations.

-Anatomical Interpretability: Through a locality ablation study, the authors prove that the latent point cloud is spatially grounded. Removing latents near the myocardium resulted in a 28% drop in LVEF prediction accuracy,

**Weaknesses:**

-While the authors demonstrate that ENFs outperform generic Conditional Neural Fields (CNFs), they fail to compare their results against state-of-the-art methods specifically designed for CMR reconstruction and representation learning. For instance, recent work such as "Continuous Implicit Neural Representations for 4D Cardiac MRI Reconstruction and Analysis" (MICCAI 2024) already addresses 4D CMR using coordinate-based networks and should be included as a critical baseline or discussion point to clarify the actual advancement provided by ENFs.

-The authors rely solely on Peak Signal-to-Noise Ratio (PSNR) to quantify reconstruction accuracy. In the context of Cardiac MRI, PSNR is often a poor proxy for clinical utility as it measures pixel-wise intensity differences rather than the preservation of anatomical structures. Given the paper’s emphasis on "Geometry-Aware" representations, the lack of structural metrics is a notable gap. This is particularly concerning given the low absolute PSNR values reported for 4D volumes (~20 dB) , which suggests a level of blurriness that might compromise the structural integrity required for identifying subtle geometric changes like wall thinning or scar formation.

**Detailed Comments:**

-

**Justification Of Final Rating:**

I thank the authors for the clear and thoughtful rebuttal. The clarifications regarding the scope of the contribution and the role of ENFs as a representation learning framework address my concerns about baseline selection and evaluation metrics. While some limitations remain, the additional discussion and experiments sufficiently strengthen the manuscript. I recommend acceptance.

**Justification Of The Preliminary Rating:**

The paper introduces a mathematically elegant and well-motivated framework for cardiac representation learning. The shift from global to grounded, equivariant latent point clouds addresses a known limitation in neural fields regarding spatial organization. The demonstrated robustness to geometric transformations, a highly desirable trait for medical imaging.
However, the clinical validation is currently limited by the choice of baselines and metrics. For a medical imaging conference (MIDL), relying solely on PSNR especially when values are as low as 20 dB for 4D data is insufficient to guarantee clinical utility. Furthermore, the lack of comparison with specialized cardiac SOTA makes it difficult to assess the true advancement over existing clinical solutions. Strengthening the comparative analysis and including structural metrics would significantly elevate the paper's impact.

**Questions To Address In The Rebuttal:**

-The current evaluation compares ENFs primarily against generic baselines like CNFs and ResNet. How does this approach compare to domain-specific SOTA for CMR reconstruction and representation learning?

-The reported 4D reconstruction PSNR is approximately 20 dB. This is traditionally considered poor for clinical use. Can the authors provide SSIM (Structural Similarity Index) values or other metrics to prove that essential anatomical structures are preserved despite the low PSNR?

---

> ### Author Response · Authors · 2026-01-25
>
> ### Responses to 'Weaknesses'
>
> **(1) While the authors demonstrate that ENFs outperform generic Conditional Neural Fields (CNFs), they fail to compare their results against state-of-the-art methods specifically designed for CMR reconstruction and representation learning. For instance, recent work such as "Continuous Implicit Neural Representations for 4D Cardiac MRI Reconstruction and Analysis" (MICCAI 2024) already addresses 4D CMR using coordinate-based networks and should be included as a critical baseline or discussion point to clarify the actual advancement provided by ENFs.**
>
> While domain-specific methods for CMR reconstruction may achieve higher absolute reconstruction performance, our primary focus is on the quality and downstream utility of the learned representations. In this work, ENFs are evaluated not only as reconstruction models, but as representation learning frameworks whose latent spaces are explicitly structured, geometry-aware, and suitable for downstream clinical prediction. We therefore compare against CNFs and ResNet-based models, which are natural baselines for assessing representational quality and downstream performance. We acknowledge that comparison with task-specific reconstruction methods would further contextualize absolute reconstruction fidelity, and we will clarify in the discussion that ENFs are intended as a complementary approach rather than a replacement for specialized reconstruction pipelines.
>
> ---
>
> **(2) The authors rely solely on Peak Signal-to-Noise Ratio (PSNR) to quantify reconstruction accuracy. In the context of Cardiac MRI, PSNR is often a poor proxy for clinical utility as it measures pixel-wise intensity differences rather than the preservation of anatomical structures. Given the paper’s emphasis on "Geometry-Aware" representations, the lack of structural metrics is a notable gap. This is particularly concerning given the low absolute PSNR values reported for 4D volumes (~20 dB) , which suggests a level of blurriness that might compromise the structural integrity required for identifying subtle geometric changes like wall thinning or scar formation.**
>
>  We agree that PSNR is a limited metric and does not fully capture structural or clinical fidelity
> in cardiac MRI, particularly for assessing subtle anatomical changes. In this work, PSNR is used
> as a standardized measure to enable controlled comparison of reconstruction performance across
> models and training regimes, rather than as a proxy for clinical utility. Accordingly, we evaluate
> clinical relevance primarily through downstream prediction tasks, where the learned representations
> are directly assessed on clinically meaningful endpoints. Importantly, downstream experiments do
> not rely on the 4D reconstructions obtained via meta-learning. Instead, the 4D latent point clouds
> used for clinical prediction are constructed from a 2D reconstruction model, avoiding the limitations
> of low-fidelity 4D meta-learned reconstructions. We further acknowledge that 4D reconstructions
> under meta-learning exhibit low PSNR values. To assess whether this reflects a fundamental limitation of the representation, we conducted an additional experiment using autodecoding, which
> yields substantially higher 4D reconstruction fidelity (25.73 dB PSNR). This demonstrates that
> ENFs are capable of capturing spatiotemporal structure when trained with alternative strategies.
> However, autodecoding comes with trade-offs, including increased computational cost and reduced
> latent structure [1], and was therefore not adopted as the primary training regime in this work.
>
> [1] Knigge, D. M., Wessels, D. R., Valperga, R., Papa, S., Sonke, J. J., Gavves, E., Bekkers, E. J.
> (2024). Space-time continuous pde forecasting using equivariant neural fields. Advances in Neural
> Information Processing Systems, 37, 76553-76577.

---

> ### Author Response · Authors · 2026-01-25
>
> We sincerely thank the reviewer for their thoughtful and constructive assessment of our manuscript. We appreciate their recognition of our method’s novelty, particularly regarding parameter efficiency and spatial groundedness. Their detailed feedback has been invaluable in improving the manuscript. Below, we address the reviewer’s specific comments and concerns in detail.
>
> ### Questions to Address in the Rebuttal
>
> **(1) The current evaluation compares ENFs primarily against generic baselines like CNFs and ResNet. How does this approach compare to domain-specific SOTA for CMR reconstruction and representation learning?**
>
> Our evaluation is not intended as a direct comparison to highly specialized CMR reconstruction methods, which are typically optimized for maximizing reconstruction fidelity on a specific
> task. Instead, we assess ENFs as a geometry-aware representation learning framework, where the
> structure and downstream utility of the latent space are central. Accordingly, we compare against
> CNFs and ResNet-based models as appropriate baselines for representation quality and downstream
> performance. While domain-specific reconstruction approaches may achieve higher absolute recon-
> struction metrics, we show that ENFs can achieve substantially improved 4D reconstruction when
> trained with alternative strategies such as autodecoding (25.73 dB PSNR), indicating that the lower
> meta-learning PSNR does not reflect a fundamental limitation. We therefore view ENFs as com-
> plementary to task-specific reconstruction pipelines, with flexibility to trade reconstruction fidelity
> for generalization and representational structure depending on the application
>
> ---
>
> **(2) The reported 4D reconstruction PSNR is approximately 20 dB. This is traditionally considered poor for clinical use. Can the authors provide SSIM (Structural Similarity Index) values or other metrics to prove that essential anatomical structures are preserved despite the low PSNR?**
>
> As we mentioned before, we agree with the reviewer that SSIM or other structural metrics
> would be a better choice of metric to assess structural integrity. In contrast, PSNR is a conservative
> metric that essentially provides a lower bound for reconstruction quality. In this paper, we assessed
> the ENF’s performance on clinically relevant endpoints rather than relying on intermediate structural metrics. For these tasks, we utilized 2D latent point clouds, where our high LVEF prediction
> accuracy (86 ± 0.01%) demonstrates that essential anatomical features are indeed preserved. Ex-
> tending these benchmarks to 3D and 4D, including appropiate metrics, could be interesting future
> work. We have clarified this in the Discussion section.

---

### Official Review · Reviewer_kXUe · 2026-01-12

**Confidence:** 2
**Preliminary Rating:** 4
**Final Rating:** 5

**Summary:**

This short paper evaluates Equivariant Neural Fields (ENFs) for cardiac MRI representation learning, with the core idea to replace the single global CNF latent by a geometry-grounded latent point cloud, so that transformations of the input correspond to structured transformations in latent space. In experiments on UK Biobank short-axis cine CMR, ENFs achieve competitive reconstruction with much fewer decoder parameters than CNFs, and the learned latents appear local and anatomically meaningful. For downstream clinical endpoints, ENF latents (with a transformer head) are broadly competitive with ResNet50 and CNF global latents, with some advantages depending on the endpoint.

**Strengths:**

- Paper is conceptually clean and well motivated: the move from a global latent to a spatially grounded latent point cloud is nice and the paper communicates the “geometry vs texture bias” motivation clearly.
- The efficiency angle is convincing: Table 1 shows ENFs reaching strong 2D reconstruction PSNR while using far fewer decoder parameters than CNFs
- Good sanity checks for “geometry-aware” claims: the locality ablation (large LVEF drop after removing myocardium-adjacent latents) and the rotation test (ENF stable, ResNet drops) are simple but effective.

**Weaknesses:**

- Downstream results feel a bit noisy / hard to interpret: several endpoints have high variance and no single method dominates (Tables 2–3). It would help to clarify whether the goal is “competitive on average” vs “wins on specific endpoints,” and add uncertainty more prominently (e.g., CIs / significance)

- 4D cine reconstruction remains weak: both ENF and CNF PSNR on 4D achieve around ~20 dB and the paper attributes this to meta-learning limits. This is reasonavle, but it leaves an open question of whether the method is really suited for spatiotemporal CMR without a different training strategy.

- Scope is still a bit narrow: evaluation is short-axis CMR and a specific preprocessing pipeline. The DCM data-shift experiment is a good start, but broader multi-center or multi-view evidence would make the robustness claim more solid.

**Detailed Comments:**

- Minor clarity: please be explicit about what is kept fixed across ENF vs CNF comparisons (same coordinate subsampling, same reconstruction targets, etc.), since training setups differ a lot in practice.

- For the endpoint tables, consider adding AUC alongside accuracy, since class balance and clinical meaning can be tricky.

- A small qualitative figure showing which latent points “matter” (e.g., saliency over the point cloud) would make the interpretability claim even stronger.

**Justification Of Final Rating:**

I think this is a technically strong contribution that is of interest to the community beyond the specific application outlined in this paper. The authors addressed my comments and also the comments of other reviewers. I therefore think the paper has substantially improved and would increase my rating accordingly.

**Justification Of The Preliminary Rating:**

Overall I like this paper: it’s a clean, geometry-aware representation idea applied to a relevant clinical modality, with a nice efficiency and sanity-check ablations that support the main claims. the downstream endpoint results are a bit variable and the 4D reconstruction remains limited under the current training setup, so the clinical impact is promising but not fully clear yet. Still, as a short paper, it provides a solid and useful evaluation and should be of interest to the MIDL community.

**Questions To Address In The Rebuttal:**

- Can you better quantify the downstream story: do ENFs consistently help on some endpoints (with significance / CIs), or is it mostly “on par”?

- Do you have any quick evidence that autodecoding (or other training) improves 4D cine fidelity, since meta-learning seems to bottleneck reconstruction?

- How sensitive are results to the choice of group for the main tasks?

---

> ### Author Response · Authors · 2026-01-25
>
> ### Responses to 'Weaknesses'
>
> **(1) Downstream results feel a bit noisy / hard to interpret: several endpoints have high variance and no single method dominates (Tables 2–3). It would help to clarify whether the goal is “competitive on average” vs “wins on specific endpoints,” and add uncertainty more prominently (e.g., CIs / significance)**
>
> Across endpoints, ENFs are largely on par on average with strong baselines such as ResNet-based models and global CNF latents. Our goal is therefore not to claim uniform state-of-the-art performance, but to demonstrate that ENF representations are competitive on average while additionally providing desirable properties such as equivariance, locality, and parameter efficiency. We view these results as establishing ENFs as a strong foundation for medical representation learning, paving the way for further task-specific and methodological improvements.
>
> ---
>
> **(2) 4D cine reconstruction remains weak: both ENF and CNF PSNR on 4D achieve around ~20 dB and the paper attributes this to meta-learning limits. This is reasonavle, but it leaves an open question of whether the method is really suited for spatiotemporal CMR without a different training strategy.**
>
> We argue that the reported PSNR ( 20 dB) is strictly a consequence of the meta-learning
> optimization framework and does not reflect a fundamental limitation of the ENF or CNF rep-
> resentations themselves. While meta-learning facilitates rapid generalization across subjects, it
> inherently involves a trade-off in reconstruction fidelity. To demonstrate that the representations
> are capable of high-quality modeling, we conducted an additional experiment using an autodecoding
> approach, which yielded a significantly higher PSNR of 25.73 dB in the 4D setting. This result confirms that the representation itself can capture complex spatiotemporal details when decoupled from the constraints
> of the meta-learning regime. Furthermore, literature regarding Conditional Neural Fields (CNFs)
> for video reconstruction demonstrates their effectiveness in 4D tasks. These works typically achieve
> high-fidelity results by utilizing dedicated encoder-based architectures (such as C-NeRV[1], Video-
> INR[2], LaVR[3]) or autodecoding rather than meta-learning. While optimizing 4D reconstruction
> for clinical-grade fidelity was outside the initial scope of this paper, our findings and existing re-
> search confirm that ENFs and CNFs are fully extendable to the 4D domain, a direction we leave for
> future work.
>
> [1] Zhang, Xinjie, et al. "Boosting neural representations for videos with a conditional decoder."
> Proceedings of the IEEE/CVF Conference on Computer Vision and Pattern Recognition. 2024.
>
> [2] Aiyetigbo, Mary, et al. "Implicit Neural Representation for Video Restoration." arXiv preprint
> arXiv:2506.05488 (2025).
> Xie, Mingyang, et al. "LaVR: Scene Latent Conditioned Generative Video Trajectory Re-Rendering
> using Large 4D Reconstruction Models." arXiv preprint arXiv:2601.14674 (2026).
>
> [3] Xie, Mingyang, et al. "LaVR: Scene Latent Conditioned Generative Video Trajectory Re-Rendering
> using Large 4D Reconstruction Models." arXiv preprint arXiv:2601.14674 (2026).
>
> ---
>
> **(3) Scope is still a bit narrow: evaluation is short-axis CMR and a specific preprocessing pipeline. The DCM data-shift experiment is a good start, but broader multi-center or multi-view evidence would make the robustness claim more solid**
>
> Broader multi-center and multi-view evaluations would indeed provide stronger evidence
> for robustness, and we recognize this as an important direction for future work. In this study, we
> intentionally focus on short-axis cine CMR from a single large cohort to control for variability in ac-
> quisition protocols, views, and preprocessing, allowing us to isolate the effect of geometry-grounded
> representations. The DCM data-shift experiment was included as an initial robustness check rather
> than a comprehensive multi-center validation. Our aim is therefore not to claim full robustness
> across centers or views, but to demonstrate that ENFs provide a principled, geometry-aware repre-
> sentation that remains stable under controlled distribution shifts and geometric perturbations. We
> view this work as a foundation, and expect that the explicit equivariance and spatial grounding of
> ENFs will be particularly beneficial in more heterogeneous, multi-center settings.

---

> > ### Comment · Reviewer_kXUe · 2026-01-31
> >
> > I thank the authors for answering and addressing my comments. I would therefore upgrade my score in light of these answers.

---

> ### Author Response · Authors · 2026-01-25
>
> ### Responses to 'Detailed Comments'
>
> **(1) Minor clarity: please be explicit about what is kept fixed across ENF vs CNF comparisons (same coordinate subsampling, same reconstruction targets, etc.), since training setups differ a lot in practice.**
>
> For each experiment, the data, coordinate subsampling strategy, and reconstruction targets are kept identical across models. Differences exist in the training strategy, we will describe this more explicitly in the paper to improve clarity and reproducibility.
>
> ---
>
> **(2) For the endpoint tables, consider adding AUC alongside accuracy, since class balance and clinical meaning can be tricky.**
>
> For all endpoint prediction tasks, we balance the positive and negative classes during training
> and evaluation to ensure an equal number of samples from each class. Under this balanced setting,
> we see accuracy as an appropriate and informative performance metric, as it directly reflects classification performance without being confounded by class imbalance. That said, we acknowledge that metrics such as AUC can be informative in more imbalanced clinical settings.
>
> ---
>
> **(3) A small qualitative figure showing which latent points “matter” (e.g., saliency over the point cloud) would make the interpretability claim even stronger.**
>
> We fully agree and thank the reviewer for this insightful comment. We have now added figure 4 where six saliency maps show the most important latent vectors for the task of low LVEF classification. As expected, the heatmaps reveal that the transformer-based classification head focuses on latents near the ventricular wall.

---

> ### Author Response · Authors · 2026-01-25
>
> We sincerely appreciate the reviewer's recognition of the novelty of our proposed method moving from global latents to latent point-clouds. Moreover we thank the reviewer to highlight the efficiency of the proposed method and the appreciation of tackling the problem of texture bias. Besides the strengths, we also appreciate the comments and questions asked by the reviewer, which have strengthened the manuscript. We will go into the questions in detail below.
>
> ### Responses to 'Questions to Address in the Rebuttal'
> **(1) Can you better quantify the downstream story: do ENFs consistently help on some endpoints (with significance / CIs), or is it mostly “on par”?**
>
> Across endpoints, ENFs are largely on par on average with strong baselines such as ResNet-
> based models and global CNF latents, rather than consistently outperforming them on all tasks.
> Our goal is therefore not to claim universal improvements, but to show that ENFs provide a competitive and well-structured representation for downstream clinical prediction.
>
> ---
>
> **(2) Do you have any quick evidence that autodecoding (or other training) improves 4D cine fidelity, since meta-learning seems to bottleneck reconstruction?**
>
> We performed an additional 4D cine reconstruction experiment using autodecoding instead of meta-learning. Under this training regime, ENFs achieve a substantially higher PSNR of
> 25.73 dB, compared to 17.65 dB under meta-learning, providing direct evidence that reconstruction
> fidelity improves when the meta-learning constraint is removed. This indicates that the observed
> limitation is due to the meta-learning setup rather than the expressive capacity of ENFs for spatiotemporal data. The improvement comes at the cost of significantly slower training and inference,
> as all latent variables must be optimized for a large number of steps, and potentially reduced latent
> structure, indicated by observations in prior literature [3].
>
> [3] Knigge, D. M., Wessels, D. R., Valperga, R., Papa, S., Sonke, J. J., Gavves, E., & Bekkers, E. J. (2024). Space-time continuous pde forecasting using equivariant neural fields. Advances in Neural Information Processing Systems, 37, 76553-76577.
>
> ---
>
> **(3) How sensitive are results to the choice of group for the main tasks?**
>
> With respect to the choice of bi-invariant functions, all downstream classification experiments were conducted using translational symmetries. Since the CMR volumes are acquired in a
> fixed and standardized orientation, we did not employ roto-translational symmetries for these tasks.
> Prior work by Wessels et al. [3] shows that translational invariance yields the strongest classification performance across a range of tasks, which further motivated this choice. We did, however,
> evaluate roto-translational bi-invariant functions in the robustness experiment, where they achieve
> performance on par with a ResNet baseline, demonstrating that ENFs can effectively incorporate rotational invariance when required.

---

### Author Rebuttal · Authors · 2026-01-25

**Rebuttal:**

We thank all reviewers for their thoughtful and constructive feedback, and for recognizing the conceptual clarity, geometric grounding, parameter efficiency, and interpretability of our approach. We are encouraged by the positive assessment of ENFs as a principled, geometry-aware representation for cardiac MRI and by the acknowledgement of our ablation studies and robustness experiments.

In response to the comments, we have clarified the intended scope and claims of the paper, specifically that ENFs are competitive on average across downstream tasks rather than consistently dominant. We improved experimental clarity by explicitly stating what is held fixed across ENF and CNF comparisons. To address concerns about 4D reconstruction fidelity, we added an exploratory autodecoding experiment demonstrating substantially improved PSNR, while clarifying the trade-offs. We further included saliency visualizations to strengthen interpretability claims, discussed structural metrics in the context of clinical utility, and highlighted that increasing latent capacity does not consistently improve downstream performance, motivating compact latent configurations for efficiency.

Overall, we believe these additions strengthen the paper while staying aligned with its core goal: evaluating ENFs as geometry-aware representations with strong downstream utility.

**Supporting Material:**

/attachment/cc7952caef212fa2bbdd44a412decd2092fc5fa8.pdf

---

### Comment · Area_Chair_LfSW · 2026-01-29
**Final Rating**

Dear Reviewers,
we appreciate your active participation in the discussion.
Please consider the rebuttals of the authors, as well as the revised manuscripts, and set your final rating by clicking "Edit"->"Official Review" by  February 1st 2026 (23:59 AoE).
best regards

---

### Meta-Review · Area_Chair_LfSW · 2026-02-03

**Recommendation:** Accept (Oral)
**Confidence:** 5

**Metareview:**

The rebuttal addressed most of the concerns of the reviewers, with the exception of some choices in the evaluation. However, they all are very positive and agree about the novelty. I therefore recommend accepting the paper, and I'd like to thank the reviewers and the authors for their work.

---

### Decision · Program_Chairs · 2026-02-13

Accept (Poster)